# Rocking Devices and the Role of Vestibular Stimulation on Sleep—A Systematic Review

**Abimanju Subramaniam *** , **Aleksandra K. Eberhard-Moscicka, Matthias Ertl** and **Fred W. Mast ***

Department of Psychology, University of Bern, 3007 Bern, Switzerland; aleksandra.eberhard@unibe.ch (A.K.E.-M.)
* Correspondence: abimanju.subramaniam@unibe.ch (A.S.); fred.mast@unibe.ch (F.W.M.)

**Abstract:** Rocking devices are widely used across different age groups to facilitate sleep. This review discusses the current literature on rocking devices and how passive vestibular stimulation influences sleep architecture, sleep oscillations, and cognitive performance. We included eight studies that conducted research with rocking devices in humans (7) and mice (1) during daytime naps and/or nighttime sleep, respectively. Overall, vestibular stimulation during sleep induced faster sleep onset, coupled with more N2 in daytime naps or N3 in nighttime sleep. Vestibular stimulation also led to more sleep spindles and better memory consolidation. Optimal stimulation intensity was around 25 cm/s$^2$, and lower intensities led to smaller effects. The findings suggest a sweet spot for vestibular stimulation intensity, promoting deeper sleep at the cost of wakefulness or N1 sleep without compromising REM sleep. While further studies are needed to thoroughly investigate the motion parameters that drive the impact on sleep and cognitive performance, rocking devices may present a promising therapeutic tool for people with disrupted sleep patterns.

**Keywords:** sleep; nap; rocking bed; passive motion; eeg; sleep spindles; slow oscillations





## 1. Introduction

A rocking motion induced by devices such as hammocks and rocking chairs is widely used across different age groups to facilitate sleep and relaxation. Also, cradles have been used over centuries to pacify babies and help them fall asleep. A large variety of rocking devices, targeting all age groups, are commercially available (including cradles and beds with a motorized rocking motion), despite as yet relatively scarce experimental evidence for their beneficial effect on sleep and cognitive functions [1,2].

Sleep disorders, such as sleep insufficiency or impaired sleep quality, can be treated using various behavioral techniques [3–5] or via pharmacological intervention [6,7]. Cognitive behavioral therapy, for example, relies on patients' compliance, and pharmacological intervention carries the risk of adverse side effects. Different sensory interventions (e.g., acoustic, odor, and vestibular stimulation), as well as other non-invasive stimulation techniques (for a recent review, see [8]), during sleep, have been investigated with the motivation of improving sleep and developing new treatment options. This review will focus on passive vestibular stimulation during sleep and its influence on sleep and cognitive performance.

### 1.1. Why Passive Vestibular Stimulation in Sleep?

There is evidence that passive vestibular stimulation during sleep modulates respiration [9,10], sleep [11–14], and cognitive performance [12,13]. Recently, the influence of vestibular input on sleep architecture was also demonstrated in mice [15]. In this study, mice were exposed to linear translation parallel to the horizontal plane during one sleep session, while their brain activity was recorded for four consecutive days. Not only did they find that the rocking motion facilitated sleep onset and prolonged the time spent in deeper NREM sleep, but they also demonstrated that this effect was not present in mice with no functional otoliths. Hence, this compellingly shows that it is the vestibular input

that drives the changes found in the sleep EEG. The importance of vestibular information in general is not surprising, because it is the major non-visual sensory source of information for self-motion perception and the primary candidate for conveying motion-specific information to the brain. Yet, it requires more specific research to better understand how passive vestibular stimulation can influence human sleep architecture.

### 1.2. Sleep and the Vestibular System

The vestibular nuclei (VN) have mesencephalic connections to the intergeniculate lateral areas (ILA), which further connect to the supra chiasmatic nuclei (SCN [16]). It is this connection that guides the chronobiological input that the vestibular information has on sleep, along with the otolithic input that is involved in the regulation of body temperature according to the biological rhythm [17,18]. The connection with the orexinergic nuclei of the hypothalamus may be the core of how vestibular stimulation influences the transition between sleep and wakefulness. At the same time, the orexinergic nuclei may influence postural control through the vestibular nuclei [16]. It is important to point out that the vestibular system has multiple projections to cortical and subcortical regions, and the overall extent of its interface with sleep cannot be covered exhaustively here.

### 1.3. Sleep, Memory Consolidation, and the Vestibular System

Sleep improves cognitive performance [19], especially in memory tasks [20–22]. Given that slow oscillations and sleep spindles have a positive effect on memory consolidation, boosting these oscillations could improve memory consolidation, which may be further facilitated via the vestibulo-hippocampal network in the brain [23–25]. Thus, passive vestibular stimulation protocols could facilitate sleep through targeting slow oscillations and sleep spindles, which has been previously demonstrated for sleep spindles by Perrault et al. [13].

### 1.4. Sleep under Minimal Vestibular Stimulation

Given the evidence that passive vestibular stimulation can influence sleep [11,13,15], it is interesting to look at conditions under which vestibular cues are altered. In microgravity, which decreases vestibular input, astronauts' total sleep time is reduced [26]. In space stations built prior to the International Space Station (ISS), light and noise pollution, as well as non-existent privacy, affected the astronauts' circadian rhythm. Nevertheless, despite engineering improvements with the ISS, the total sleep time and the proportion of REM sleep did not increase, indicating that microgravity may affect sleep [26]. Interestingly though, astronauts do not seem to show signs of sleep deprivation such as reduced alertness or memory impairment. It is important to mention that the effect of microgravity on sleep could be mediated by other changes in body processes (e.g., cardiovascular [27]). Moreover, a reduction in REM sleep was also reported in other isolated confined environments (e.g., in the Antarctic or in submarines [28]), further complicating a straightforward interpretation of the effect of microgravity on sleep.

## 2. Methods

This systematic review was conducted according to the PRISMA 2020 guidelines [29]. The keywords run in the databases and search engine www.scholar.google.com (accessed on 17 October 2023) were: "rocking", "bed", "EEG", and "sleep", as well as "motion", "EEG", and "sleep" without further limitations (see also Figure 1). In both searches, the keywords were connected using the "AND" term. Two independent reviewers (A.S. and A.K.E.-M.) screened the records, selected the papers, and collected the data. Only empirical studies with humans/animals exposed to mechanical passive vestibular stimulation that measured sleep EEG and analyzed sleep macro- and micro-architecture, or macro-architecture only, were considered. Additionally, for human studies, sleep had to be scored according to AASM criteria. Also, papers concerning the technical setup but not the sleep EEG as such e.g., [30,31] were excluded from the selection. The databases used were APA PsycINFO,

PSYINDEXplus Literature and Audiovisual Media, Pubmed, and Scopus accessed on 17 October 2023.

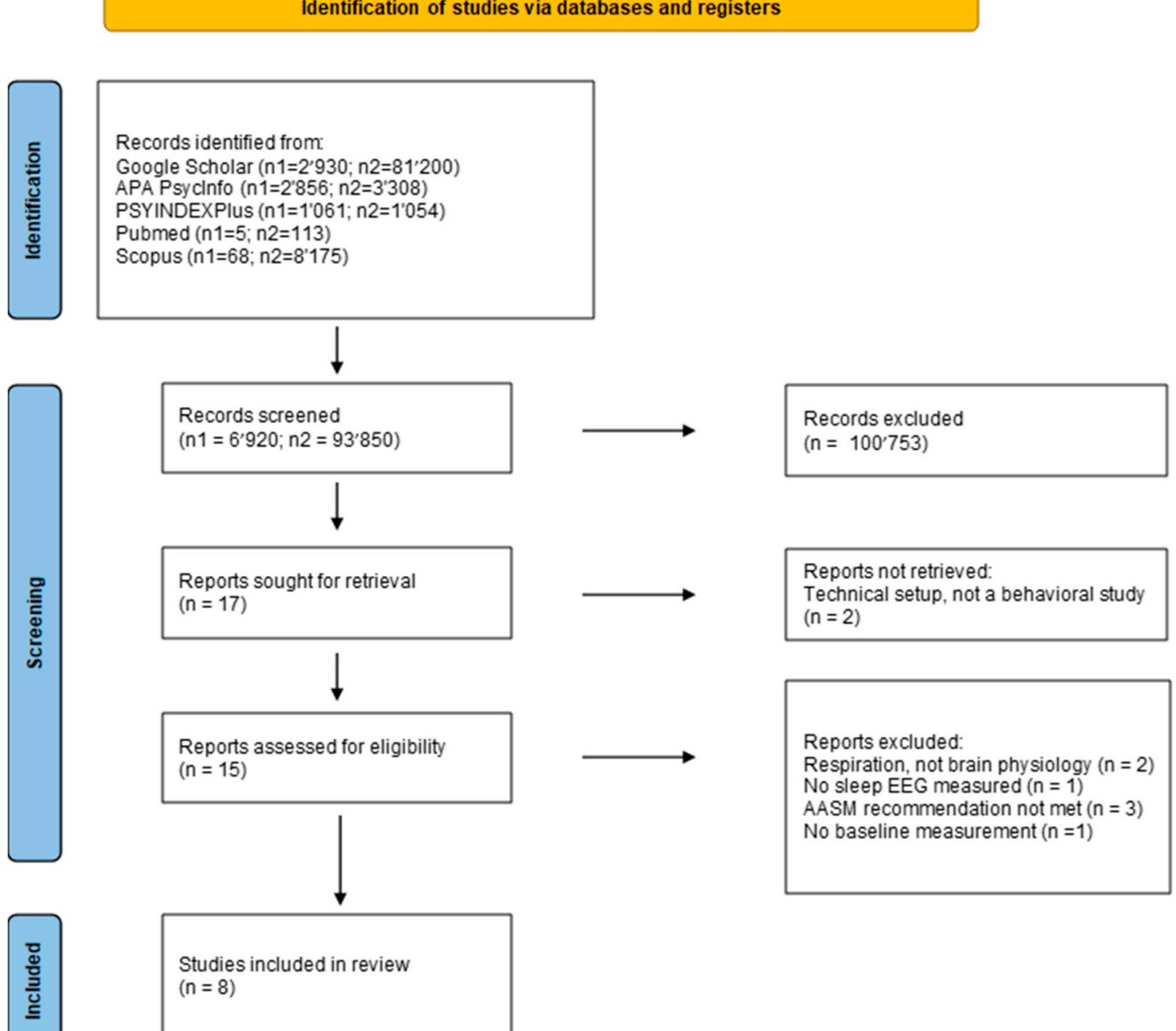

**Figure 1.** Flow diagram of the systematic search, visualizing the reports that were identified and included in the current review. Reports assessed for eligibility were further descriptively categorized based on the implemented methodology (Supplementary Table S1). n1 = search with keywords: "rocking AND bed AND EEG AND sleep"; n2 = search with keywords: "motion AND sleep AND EEG".

We used the terminology from Ertl et al. [32] to describe the vestibular motions applied to the head (i.e., rotation as pitch, yaw, or roll, and translation in inter-aural, head-vertical, or naso-occipital axes), with the assumption that participants were in a supine position.

### 3. Studies

Eight papers fulfilled the search criteria (Table 1; means and SDs of all the reported outcomes, and the mean differences, along with the calculated effects sizes for the significant differences of the reported outcomes, are reported in Supplementary Tables S2–S5). Seven papers were not considered due to the reasons detailed below. Two papers focused on respiration and not on brain physiology [9,10]. The study by Ashida et al. [33] was excluded

as it measured the subjectively perceived effectiveness of the sleep-inducing effect without recording sleep EEG. Three further studies were discarded [34–36] since they did not use the minimum number of electrodes for each recommended region (i.e., frontal, central, and occipital) that are necessary to score sleep according to AASM criteria. Moreover, in the study by Muto et al. [34], participants triggered the rocking condition themselves, leading to a lack of blinding and possibly to placebo effects. One rodent study was also excluded [37] due to the lack of a baseline measurement.

**Table 1.** Summary of studies selected with a rocking device paradigm.

| Paper | Sleep Type | Sample | Motion Type | | Independent Variables | | | Changes in | | |
|---|---|---|---|---|---|---|---|---|---|---|
| | | N | Linear Intensity in cm/s$^2$ | Angular Intensity in °/s$^2$ | Conditions | Condition Types | Stimulus Axes | SMA | SMI | CP |
| Woodward et al. (1990) [38] | Night | 8 Men | 21.6 | NA | 2 | Motion | HV | − | NA | NA |
| | Nap | 7 Men | 21.6 | NA | 2 | Motion | HV | 0 | NA | NA |
| Bayer et al. (2011) [11] | Nap | 10 Men | 25.9 | NA | 2 | Motion | IA | + | + | NA |
| Shibagaki et al. (2017) [39] | Nap | 7 Men | 21.3 | NA | 3 [4] | Motion or Aroma | HV & NO | + | NA | NA |
| Omlin et al. (2018) [12] | Night | 18 Men | 10.1/15 | NA [3] | 3 | Timing of Motion | HV/IA/NO/IA & Yaw/HV & Pitch | + [5] | + [5] | 0 |
| Kompotis et al. (2019) [15] [1] | Night [2] | 16 Mice | 4.9, 19.7, 79 & 177.7 | NA | 5 | Frequency | Horizontal translation | + | + | NA |
| | Night [2] | 9 Mice | 31.6, 79 & 177.7 | NA | 4 | Amplitude | Horizontal translation | 0 | 0 | NA |
| Perrault et al. (2019) [13] | Night | 10 Women 8 Men | 25.9 | NA | 2 | Motion | IA | + | + | + |
| van Sluijs et al. (2020a) [14] | Nap | 22 Men | 15, 25 & 35 | NA [3] | 4 | Intensity | IA & Yaw | + | + | 0 |
| Van Sluijs et al. (2020b) [40] | Night | 8 Women 11 Men | 15 | 2.15 | 2 | Motion | HV & Pitch/IA & Yaw | 0 | + | 0 |

SMA = sleep macro-architecture, SMI = sleep micro-architecture, CP = cognitive performance; HV = Head-vertical, IA = Inter-aural, NO = Naso-occipital. + refers to a positive change, − refers to a negative change, 0 refers to no influence of the rocking device, and NA if the variable was not tested. [1] Rocking device studies with mice using two different paradigms. [2] Mice were tested during their natural sleeping time, with them being nocturnal animals. [3] Angular motion induced but intensity and amplitude not reported for angular motions. [4] Motionless combined with aroma as one condition. [5] Only for the first 2 h after sleep onset.

### 3.1. Daytime Naps on Rocking Devices

The pioneering study on rocking beds by Woodward et al. [38] initiated the research on rocking devices in 1990. In this study, participants were allowed to nap five times a day for a maximum of 10 min. However, no effects concerning the duration of sleep and sleep stages were found for the head-vertical translation condition as compared to the motionless conditions. Three decades later, Bayer et al. [11] revitalized rocking device research with their study testing human sleep on a rocking bed during a 45-min nap. They found that inter-aural translation led to faster sleep onset and longer N2 sleep, which was combined with higher sleep spindle density in N2 sleep. Similarly, Shibagaki et al. [39] reported an overall increase in deep sleep (i.e., N2 and N3 pooled together) when applying a combination of head-vertical and naso-occipital translation with a chair-like rocking device. Van Sluijs et al. [14] indicated the importance of intensity as a rocking parameter, as they found deeper sleep and higher delta activity when applying a combination of yaw rotation and inter-aural translation with an intensity that was similar to the one used by Bayer et al. [11]. This study also assessed cognitive performance in the different conditions (i.e., motion and motionless naps), but did not report any differences in memory tasks.

### 3.2. Nighttime Sleep on Rocking Devices

Woodward et al. [38] also tested night sleep (i.e., two consecutive nights with head-vertical translation and two consecutive motionless nights) and found shorter N2 sleep in the head-vertical translation compared to the motionless control condition. Omlin et al. [12] conducted a study in which the participants tried head-vertical, naso-occipital, and inter-aural translation and yaw rotation combined with inter-aural translation, and pitch rotation combined with head-vertical translation, on the bed before choosing their preferred single-axis rocking motion for their sleep sessions. They were then stimulated with their preferred type of motion either until sleep onset or for the first 2 h after sleep onset. They reported changes in sleep architecture, such as more N2 sleep and an increase in sleep spindles, but, importantly, only for the first 2 h while the participants were rocked, indicating that, after 2 h of rocking, the effect dissipated, and participants slept similarly in the motion and motionless conditions. Perrault et al. [13] found that inter-aural translation during the entire night leads to shortened sleep onset, more N3 sleep resulting from less N1 and N2 sleep, and a higher sleep spindle density. In the same study, they also found improved memory consolidation in the morning after the translation condition when compared to the motionless condition. Conversely, a study with older people (60–75 years) did not find any differences in a declarative memory task and sleep macro-architecture but reported a reduced delta power in translation combined with rotation (head-vertical and pitch, or inter-aural and yaw) as compared to the motionless condition [40].

### 3.3. The Effect of Stimulus Intensity

Overall, shorter sleep onset during rocking conditions was reported across the literature, which either led to more N2 sleep or more N3 sleep. In some studies, REM sleep was negatively affected by the rocking motion but in most instances, it was not. Mice, for example, had more deep sleep at the cost of REM sleep when they were rocked with a higher intensity [15]. A closer investigation of the maximal intensities of the rocking motions reveals why some studies might have found differences in N2 or N3 sleep, while others did not. While a maximal linear intensity of 22–26 cm/s$^2$ [11,13,14] altered human sleep architecture, an intensity of 15 cm/s$^2$ or lower [12,14] was less effective. According to the available literature, it is likely that an intensity of 15 cm/s$^2$ is below the vestibular perception threshold [41] (In frequencies below 1 Hz, vestibular sensitivity is decreasing [42], e.g., for 0.2 Hz, the threshold for inter-aural translation is estimated to be around 16.3 cm/s$^2$ [43]. In Omlin et al. [12], 12 participants were stimulated with 0.16 Hz and 6 participants with 0.24 Hz; as such, the used intensity was below or close to the perceptual threshold for the applied motion profiles), possibly leading to a lack of effects. The fact that mice need a 3–4 times higher intensity for sleep to be affected is noteworthy [15], as mice have 3–4 times less sensitive vestibular afferents than humans [44,45]. This suggests that the intensity of the motion stimulus matters, and it appears as a decisive factor of rocking device interventions. Notably, all the studies discussed above were rather underpowered, with a maximum of 22, mostly male, human participants. More research is therefore needed with larger and more representative samples to draw more generalizable conclusions.

### 4. Discussion

Taken together, there is compelling evidence for rocking devices to facilitate sleep (in particular, faster sleep onset, decreased wakefulness coupled with increased N2 or N3 sleep, and more sleep spindles), and some promising indication of an influence of rocking devices on memory performance. Depending on the type of sleep (i.e., nap or night sleep), either N2 or N3 sleep benefits from the rocking motions of the device. Likewise, a rather consistent increase in sleep spindles was reported in the rocking condition [11–15]. Despite the above-mentioned positive results, other studies reported either possibly short-lasting effects [12], null findings [40], or even reverse effects (i.e., a decrease in N2 sleep [38]). At this point in time, there still is remarkable heterogeneity in the results obtained. Nevertheless, they do not speak for the absence of an effect but rather for a profound lack of knowledge, and

the strong variation of experimental parameters between studies renders comparability difficult. Still, little is known about changes in REM sleep; REM sleep does not appear to be affected by low intensities [15,19], whereas its proportion decreased when the intensity was high [15]. The human studies are all based on small sample sizes, leading to at most a power of 0.61 for a medium effect size of f = 0.25 (i.e., 61% chance of finding a medium-sized effect) as calculated with G*Power (Version 3.1) [46]. Thus, this indicates the need for future studies with larger sample sizes.

Even though the current findings cannot be considered fully conclusive, there are solid hints that rocking devices have a positive effect on sleep. Hence, they have a promising potential to alleviate fragmented sleep, help insomnia patients, or even promote astronauts' sleep in microgravity. Once the optimal rocking settings are better understood, rocking devices could be used to facilitate faster sleep onset and reduce awakenings during the night. The current evidence indicates a sweet spot in the intensity of rocking so that there is an increase in N2 and/or N3 sleep, which is accompanied by a decrease in wakefulness and/or N1 sleep while REM sleep remains unaffected. Future studies should investigate whether the sweet spot varies between individuals and whether it changes across the lifespan.

Further aspects to be addressed are the type of motion (i.e., translation and rotation) that determines if otoliths and/or semicircular canals are being primarily stimulated, and the stimulation axes (i.e., head-vertical, inter-aural, and naso-occipital) that determine the direction of stimulation. In the studies reviewed here, all stimulation consisted either of at least one translation alone [11–13,15,38] in combination with another translation [39] or in combination with another rotation [12,14,40] (summarized in Table 2). Hence, in all those studies, otoliths were targeted by passive vestibular stimulation, while semicircular canals were not targeted separately. While translation seems to have a positive effect on sleep (i.e., faster sleep onset, deeper sleep [11,13,15]), for rotation the picture is less clear, since, in the studies involving rotation, participants chose the preferred motion, and the results were not analyzed separately [12,14,40]. Of note, the majority of the human studies stimulated the head-vertical [12,38–40] and inter-aural [11,12,14,40] axes, whereas the naso-occipital axis was rarely stimulated [12,39], and in the rodent study [15], the mice were stimulated in the horizontal plane. Inter-aural stimulation consistently led to shorter sleep onset coupled with deeper sleep [11,13] and head-vertical stimulation led to less conclusive results [12,38,39]. Rotation around the head-vertical axis (yaw [12,14,40]) and around the inter-aural axis (pitch [12,40]) also did not yield conclusive results. Furthermore, given that the reviewed papers were conducted under different settings (i.e., nighttime or daytime sleep) and applied different stimulus intensities, the stimulation type and the axes used in the different studies cannot be directly compared. Also, considering that participants had different preferences with respect to different motion types and stimulation axes [12,40], future studies may further investigate if and how the preference of the motion affects sleep.

Given the bidirectional connection between the vestibular and hypothalamic areas [16], it is possible that there is an effect in the reverse direction. Rocking devices can also lead to better vestibular functions through the exposure to passive vestibular stimulation and can be moderated via the improvement in sleep quality. For example, lower vestibular-evoked myogenic potentials (VEMP) were associated with Alzheimer's disease, indicating a link between the vestibular and cognitive part of the brain [47], even though this does not imply causality. Nevertheless, if higher passive vestibular stimulation intensities yield better effects, the impact of passive vestibular stimulation during sleep on hippocampal volume in people at risk of developing Alzheimer's disease would be worth investigating. In this sense, future research with rocking devices could be used to improve cognitive performance [13]. While not all types of memory are expected to benefit from passive vestibular stimulation (e.g., procedural memory), declarative memory, such as episodic memory, as well as semantic memory, could profit in the long term, given that the vestibulo-hippocampal network facilitates memory consolidation [26–28]. Last but not least, the lack of changes in sleep architecture in older people found by van Sluijs et al. [40] could be due

to aging-related vestibular sensory loss [42]. A decline in sensory processing will reduce the afferent signal strength, thus making it harder to alter sleep architecture.

Taken together, the use of rocking devices is promising in the context of sleep and its alterations, but the potential for other applications of this sensory stimulation is not yet exploited, deeming it particularly interesting to extend this line of research.

**Table 2.** Summary of the studies with the given rocking device paradigm and the induced effects on sleep macro- and micro-architecture.

| Stimulus Axes | Paper | Sleep Macro-Architecture | | | | | | | Sleep Micro-Architecture | | | | |
|---|---|---|---|---|---|---|---|---|---|---|---|---|---|
| | | SO | N1 | N2 | N3 | REM | W | SE | Delta Power | #SSO | SSO Density | #SS | SS Density |
| Horizontal translation | Kompotis et al. (2019) [15] [1] | − | NREM [4]: + | | | − [9] | − | NA | − [9] | NA | NA | − [9] | − [9] |
| Inter-Aural | Bayer et al. (2011) [11] [2] | − | − | + | 0 | NA | − [10] | 0 | + | NA | NA | + | + |
| | Perrault et al. (2019) [13] [2] | − | 0 [5] | 0 [5] | + | 0 | 0 | 0 | NA | + [11] | 0 | + [11] | + [11] |
| | Omlin et al. (2018) [12] [1,3] | 0 | 0 | + [6] | 0 | 0 | 0 | 0 | 0 | 0 | 0 | + [6] | 0 |
| Inter-Aural × Yaw | Van Sluijs et al. (2020a) [14] [2] | 0 | − | 0 | + [8] | NA | 0 | 0 | + [11] | 0 | + [8] | + | 0 |
| | Van Sluijs et al. (2020b) [40] [2,3] | 0 | 0 | 0 | 0 | 0 | 0 | 0 | − | 0 | 0 | 0 | 0 |
| | Omlin et al. (2018) [12] [1,3] | 0 | 0 | + [6] | 0 | 0 | 0 | 0 | 0 | 0 | 0 | + [6] | 0 |
| Head-Vertical | Woodward et al. (1990) [38] [2] | 0 | 0 | 0 | 0 | 0 | 0 [10] | 0 | NA | NA | NA | NA | NA |
| | | 0 | 0 | − | 0 | 0 | 0 [10] | 0 | NA | NA | NA | NA | NA |
| | Omlin et al. (2018) [12] [1,3] | 0 | 0 | + [6] | 0 | 0 | 0 | 0 | 0 | 0 | 0 | + [6] | 0 |
| Head-Vertical × Naso-Occipital | Shibagaki et al. (2017) [39] [1] | 0 | NA | 0 [7] | 0 [7] | NA | NA | NA | NA | NA | NA | NA | NA |
| Head-Vertical × Pitch | Van Sluijs et al. (2020b) [40] [2,3] | 0 | 0 | 0 | 0 | 0 | 0 | 0 | − | 0 | 0 | 0 | 0 |
| | Omlin et al. (2018) [12] [1,3] | 0 | 0 | + [6] | 0 | 0 | 0 | 0 | 0 | 0 | 0 | + [6] | 0 |
| Naso-Occipital | Omlin et al. (2018) [12] [1,3] | 0 | 0 | + [6] | 0 | 0 | 0 | 0 | 0 | 0 | 0 | + [6] | 0 |

SO = sleep onset, N1 = non-rapid eye movement sleep 1, N2 = non-rapid eye movement sleep 2, N3 = non-rapid eye movement sleep 3, REM = rapid eye movement sleep, W = wake, SE = sleep efficiency, #SSO = number of sleep slow oscillations, SSO Density = sleep slow oscillations density, #SS = number of sleep spindles, SS Density = sleep spindle density + refers to a positive change, − refers to a negative change, 0 refers to no influence of the rocking device, and NA if the variable was not tested. [1] Sleep stages reported as duration in min. [2] Sleep stages reported as percentage of total sleep time. [3] Participants were allowed to choose their preferred motion, but the authors did not provide details regarding the separate motions. [4] In rodents no differentiation within NREM sleep. [5] Decrease only when N1 and N2 pooled together. [6] Only during two hours of stimulation. [7] Increase only when N2 and N3 pooled together. [8] Only in one of the two groups tested. [9] Only when rocking with the highest intensity. [10] Number of Awakenings. [11] Only in N3.

## 5. Conclusions and Future Directions

The current literature on rocking devices is still relatively scarce, and further studies are needed, not only to thoroughly investigate the motion parameters that drive the impact on sleep, but also to strengthen the findings and to deliver stronger statistical evidence. Ultimately, rocking devices may present a promising therapeutic tool not only for patients with sleep disorders and cognitive impairments, but also for older people suffering from fragmented sleep. Rocking devices may further prove a more cost-effective treatment due

to the vast outlay of medication or time needed for a successful CBT-I, as compared to a rocking device that would need to be set up once without a need for further assistance. Moreover, to broaden the mechanistic understanding of the influence of rocking devices, patients with vestibular impairment should also be investigated.

## 6. Limitation

Since no automation tool was used to screen the records, human error may be possible. Furthermore, we did not contact study authors to seek additional information in cases where the information provided was unclear or lacking.

**Supplementary Materials:** The following supporting information can be downloaded at: https://www.mdpi.com/article/10.3390/ctn7040040/s1, Table S1. Categorization of studies based on the implemented methodology; Table S2. Mean ± SD for sleep macro-architecture outcomes; Table S3. Mean ± SD for sleep micro-architecture outcomes; Table S4. Mean differences for all the reported outcomes and Hedges' g for all the significant differences; Table S5. Mean ± SD and the mean differences for all the reported memory outcomes, along with Hedges' g for all the significant differences.

**Author Contributions:** Conceptualization, all; methodology, all; investigation, A.S.; writing—original draft preparation, A.S. and A.K.E.-M.; writing—review and editing, F.W.M. and M.E.; visualization, A.S.; supervision, F.W.M.; project administration, A.S., A.K.E.-M. and M.E.; funding acquisition, F.W.M. All authors have read and agreed to the published version of the manuscript.

**Funding:** This research was funded by the Interfaculty Research Cooperation "Decoding Sleep", and the sitem-insel support funds (SISF), both from the University of Bern.

**Data Availability Statement:** No new data were created or analyzed in this study. Data sharing is not applicable to this article.

**Acknowledgments:** We are grateful for technical support from Carlo Prelz from the Technology Platform of the Human Sciences Faculty of the University of Bern.

**Conflicts of Interest:** The authors declare no conflict of interest.

**Registration and Protocol:** This review had not been registered beforehand, and a review protocol was not prepared.

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
