# Peer review of "Rocking Devices and the Role of Vestibular Stimulation on Sleep—A Systematic Review"

_ctn, doi:10.3390/ctn7040040_

Round 1
Reviewer 1 Report
Comments and Suggestions for Authors
Authors submitted a review focused on the effects of rocking devices on sleep and, to an extent, on cognition.
The topic is surely of interest, however, I have several concerns regarding the aim of the review, the papers that were reviewed, as well as the conclusions.
General comments
- Authors stated in the title that they focused on the effects of rocking devices on sleep and cognition. However, the paper is focused on sleep and cognition is only adressed because of the role of sleep in cognitive functionning. Maybe it would be an idea to remove the term cognition in the title ?
- Seemly, it is not clear to me why authors talk about relaxation in the abstract.
- Why did authors focused on the effects of passive devices ? Then, the sentence stating "how vestibular stimulation influences sleep architecture..." is partly wrong given that authors investigated the effects of passive stimulation and not of other types of stimulation.
- Then, it would be helpful to make it clearer in the whole article that there is a focus on passive stimulation and that other type are discussed elsewhere (for example: Park et al., 2023, Biomed Eng Lett.).
- I miss a discussion related to the effects of various axes on sleep (as it has been done for the effects of stimulus intensity). Could authors add a paragraph about this ? Such a paragraph would be helpful to better understand which parts of the vestibular system are involved and what type of passive stimulations are better suited.
Specific comments
- I suggest to remove the paragraph related to microgravity that is not the purpose of the paper.
- Please update the titles 1.2 and 1.3 to make it clearer about the content (for example the 1.3 is not related to sleep and memory but to the effects of rocking on sleep and memory). Also, I am not sure there is enough papers related to this field to make one separated paragraph.
- Regarding search terms, authors choose ones that are very broad and not specifically related to the topic adressed in the review. I would suggest to include, at least, polysomnography or EEG. If authors want to focus on memory, this should be added as a keyword. Also, did authors used the terms "AND, "OR" ? Please specify. Search terms need to be adapted and more specific.
- Please include in Table 1 information related to the population (humans or rodents).
- As a side note, it would be useful to add a figure explaining the potential effects of passive vestibular stimulation on sleep.
Author Response
Please see the attachement.

Reviewer 2 Report
Comments and Suggestions for Authors
This review manuscript is well written and provides a useful summary of the influence of rocking devices and vestibular stimulation on sleep behavior. I recommend acceptance of the manuscript following one major change.
Please provide the following:
It would be of great use to any reader to have a table summarizing the different phases and/or elements of sleep that were explored. I suggest creating a table that briefly outlines all different sleep parameters, how they are quantified and what they represent. This will allow the reader to more easily understand how the different vestibular stimulation paradigms contributed to changes in each of the sleep elements measured.
Author Response
Please see the attachement.

Round 2
Reviewer 1 Report
Comments and Suggestions for Authors
Dear authors,
thank you for the revision of the manuscript. The clarity has been improved and a lot of work has been done.
I have some remarks, still. See below.
- Please remove the term "Systematic review" as the current manuscript is not a systematic review, that needs to follow specific steps (study validity, flow chart etc). See doi: 10.1258/jrsm.96.3.118.
- Thank you for udpating Table 1, however, I could nto find the terms "men" and "women". I may have msunderstood authors' answer. Table 2 brings in new elements of interest, indeed. Thank you for this additional table.
- Please correct for the title "Macroarchitecture" in this table 2 , there is a mispelling.
- In Table 2, I would not have separate sleep microarchitecture and power spectrum (both refers to microarchitecture for me). Also, the delta band is related to slow waves sleep that are related to memory consolidation. Also, please specify if information is related to sleep stages duration, percentage or something else.
- In the discussion, I would be more enthusiatic about the fact that passive vestibular stimulation improve sleep and cognition as only 8 studies were reviewed, with different methodologies. Although spindles and slow waves are involved in memory consolidation, Bayer et al., did not report memory improvement as well as van Sluijs et al., 2020. Only Perrault and colleagues reported such improvement which is not enough to definitively conclude about positive effects of rocking on memory. Also, sleep did not improve in all stages and in all studies.
Could authors please specify the percentage of studies that found improvement/reduction in sleep (N3 ; N1/N2 ; spindles ; slow waves ; delta) and cognition ? Such information would give more precise information related to the potential beneficial effects of rocking on sleep and cognition.
- Also, please inform authors that a decrease in light sleep but an increase in deep sleep is an overall positive effect (van Sluijs et al.,). However, this is not the case in all studies and I miss a clear discussion about the heterogeneity of the results.
